# SentenceKV: Efficient LLM Inference via Sentence-Level Semantic KV Caching[*]

**Yuxuan Zhu**
Department of Computer Science
Rensselaer Polytechnic Institute
Troy, NY 12180, USA
zhuy27@rpi.edu

**Ali Falahati**
Department of Computer Science
University of Waterloo
Waterloo, ON N2L 5Z5, Canada
afalahat@uwaterloo.ca

**David H. Yang**
Department of Computer Science
Rensselaer Polytechnic Institute
Troy, NY 12180, USA
yangd13@rpi.edu

**Mohammad Mohammadi Amiri**
Department of Computer Science
Rensselaer Polytechnic Institute
Troy, NY 12180, USA
mamiri@rpi.edu

## Abstract

Large language models face significant computational and memory challenges when processing long contexts. During inference, efficient management of the key-value (KV) cache, which stores intermediate activations for autoregressive generation, is critical to reducing memory overhead and improving computational efficiency. Traditional token-level efficient KV caching methods overlook semantic information, treating tokens independently without considering their semantic relationships. Meanwhile, existing semantic-preserving KV cache management approaches often suffer from substantial memory usage and high time-to-first-token. To address these limitations, we propose **SentenceKV**, a novel sentence-level semantic KV caching approach designed to enhance inference efficiency while preserving semantic coherence. During **prefilling**, SentenceKV groups tokens based on sentence-level semantic similarity, compressing sentence representations into concise semantic vectors stored directly on the GPU, while individual KV pairs are offloaded to CPU. During **decoding**, SentenceKV generates tokens by selectively retrieving semantically relevant sentence-level KV entries, leveraging the semantic similarity between the prefilling-stage semantic vectors and decoding-stage queries. This ensures efficient and contextually accurate predictions, minimizing the loading of redundant or irrelevant data into GPU memory and significantly reducing memory overhead while maintaining stable inference latency, even for extremely long contexts. Extensive evaluations on benchmarks including PG-19, LongBench, Needle-In-A-Haystack, and RULER demonstrate that SentenceKV significantly outperforms state-of-the-art methods in both efficiency and memory usage, without compromising model accuracy.

## 1 Introduction

Large language models (LLMs) have achieved remarkable success across a wide range of natural language processing tasks (Min et al., 2023; Zhao et al., 2023). As LLMs are increasingly applied to complex reasoning and multi-step decision-making scenarios, the demand for longer context windows has grown significantly Chen et al. (2023). Recent advancements have extended the context window from the early GPT models to the latest iterations, such as GPT-o1 and beyond (OpenAI, 2023; Koubaa, 2023). However, this growth

---

[*]Code available at https://github.com/zzbright1998/SentenceKV

introduces substantial challenges in key-value (KV) cache efficiency, leading to memory bottlenecks and increased inference latency Shi et al. (2024).

Due to the transformer-based architecture of LLMs Vaswani et al. (2017), the memory footprint of KV cache grows linearly with the context length, directly impacting GPU usage. For example, in the case of the LLaMA-3.1-8B model AI (2023), as shown in Appendix A.1, processing a 32k-token prompt during the decoding stage requires approximately **16** GB (using float16 precision) of GPU memory, which can be prohibitive for users without high-end hardware. Furthermore, longer prompts significantly increase computational overhead, as each additional token requires computing attention weights across the entire sequence. This quadratic attention complexity leads to substantial compute inefficiency, making efficient inference particularly challenging in resource-constrained environments.

To mitigate these issues, several methods have been proposed to compress KV cache Shi et al. (2024). Among them, token eviction strategies based on fixed selection patterns or attention-weight-based pruning have shown promise (Xiao et al., 2023; Zhang et al., 2023). However, these approaches suffer from key limitations: they fail to adapt to the dynamic nature of token importance during decoding and often discard tokens permanently, neglecting their potential relevance in the future decoding steps. This static behavior can degrade the performance of the model, especially in tasks requiring complex reasoning and long-range dependencies Liu et al. (2025a).

Another category of approaches, such as Quest Tang et al. (2024) and ShadowKV Sun et al. (2024), dynamically retrieve portions of KV cache from CPU-offloaded storage back to GPU memory during attention calculation. Typically, these methods segment the input into fixed-size chunks and selectively retrieve them based on query relevance at each decoding step. However, fixed-size chunking can disrupt semantic coherence by arbitrarily breaking natural semantic boundaries. Some dynamic retrieval approaches, including ClusterKV Liu et al. (2024) and SqueezeAttention Hooper et al. (2024), mitigate this issue by clustering semantically related keys into coherent chunks. Yet, these clustering-based methods introduce significant computational overhead, resulting in increased time-to-first-token (TTFT) latency.

To address these limitations, we propose **SentenceKV**, a novel approach that dynamically manages KV cache based on sentence-level semantic information. Our key insight is that sentences inherently encapsulate richer semantic information compared to isolated tokens, making them a more effective unit for caching and retrieval. For instance, as illustrated in Figure 1, sentences from different domains tend to form distinct semantic clusters within the representation space of LLMs, a phenomenon also observed in recent literature Chen et al. (2024). We further propose a novel multi-query-aware mechanism for retrieving relevant sentence-level KV cache entries during decoding.

SentenceKV introduces a sentence-level KV cache compression method that operates effectively during both the prefilling and decoding stages. During prefilling, our method stores the long input prompt at sentence granularity on CPU while maintaining a compact semantic representation vector on GPU. During decoding, we maintain a cache of multi-query vectors computed from the tokens of the most recently generated sentence. By averaging these vectors, we retrieve the most relevant prefilled sentences and dynamically load their corresponding KV pairs to GPU for attention computations. This approach ensures efficient recall of important tokens and preserves the semantic context, significantly reducing computational overhead while maintaining high inference efficiency.

Our method includes: (1) **Semantic token grouping** based on sentence chunking without complex clustering. (2) **Adaptive decoding cache** which maintains a dynamic cache for KV pairs retrieval aligned with coherent semantic units (e.g., full sentences) during decoding. (3) **Efficient recall** of sentence-level KV entries to maintain coherence. Compared to existing KV compression methods, SentenceKV achieves superior semantic preservation and improves recall precision efficiently.

Extensive experiments on long-context benchmarks, including LongBench Bai et al. (2023), Needle-In-A-Haystack (NIAH) Kamradt (2023), and RULER Hsieh et al. (2024) demonstrate that our approach not only improves inference efficiency but also maintains or even enhances

model accuracy across diverse tasks. Quantitative analysis shows that our method offers an efficient solution for long-context processing.

## 2 Related Work

Transformer-based models rely on KV cache for efficient inference. However, as context length increases, KV cache can consume substantial memory, imposing practical limitations on long-sequence processing. To address this, various techniques have been proposed to compress or optimize KV cache.

**KV Cache Compression.** StreamingLLM Xiao et al. (2023) enables infinite-context handling via attention sinks and a moving window, allowing models to process extremely long sequences without growing memory footprints. SnapKV Li et al. (2025) preserves essential KV positions based on per-head attention patterns with an observation window. FastGen Ge et al. (2023) progressively drops tokens with low future impact, enhancing efficiency with minimal quality loss. PyramidInfer Yang et al. (2024) and PyramidKV Cai et al. (2024) propose a layer-wise pyramidal KV cache retention pattern, preserving more tokens in upper layers of the model to maximize memory savings without sacrificing context.

**Dynamic Access KV.** Dynamic access mechanisms adaptively manage KV retention based on task-specific requirements. $H_2O$ Zhang et al. (2023) dynamically retains high-impact tokens based on accumulated attention scores. TOVA Oren et al. (2024) prunes tokens based on real-time query relevance, but risks removing context needed for future steps. The quest Tang et al. (2024) partitions KV cache into pageable memory and predicts critical pages per query, in which case fragmentation issues can arise. InfLLM Xiao et al. (2024) stores distant contexts into additional memory units and employs an efficient mechanism to lookup token-relevant units for attention computation. ShadowKV Sun et al. (2024) offloads the value cache to CPU and stores a low-rank key cache on GPU, using a KV selection strategy to reduce memory footprint and maintain high-throughput decoding. SqueezeAttention Hooper et al. (2024) optimizes cache allocation across both sequence and layer dimensions, assigning larger budgets to important layers while pruning unimportant ones. DynamicKV Zhou et al. (2024) implements task-aware adaptive retention, adjusting KV budgets per layer to maintain task-specific relevance while reducing cache size.

**Semantic-Level Optimization.** As LLMs tackle increasingly complex tasks, optimizing KV cache at a semantic level is crucial to maintaining output coherence. ChunkKV Liu et al. (2025b) groups tokens into semantic chunks, retaining the most informative segments and discarding redundant ones to enhance long-context inference efficiency. ClusterKV Liu et al. (2024) introduces semantic clustering, which, unlike methods that permanently evict tokens or recall them based solely on textual positions, clusters tokens semantically and recalls them at the granularity of semantic clusters. Task-KV He et al. (2025) differentiates between heterogeneous and general aggregation attention heads, preserving full KV cache for critical heads while reducing it for less relevant ones. SepLLM Chen et al. (2024) introduces a data-dependent sparse attention mechanism to mitigate the excessive attention allocated to seemingly uninformative separator tokens. By compressing segment-level information into these tokens, SepLLM reduces KV cache while preserving essential content.

While chunk-based compression better preserves semantics than token-wise eviction, it may still fragment full thought units, omitting crucial context. Prior studies show that ignoring natural boundaries, such as sentence breaks, can harm language modeling performance (Dai et al., 2019; Rae et al., 2019). SentenceKV addresses this by retaining entire sentences or semantically self-contained segments, ensuring holistic context preservation. By bridging local semantic retention from ChunkKV with global coherence, SentenceKV further enhances long-context inference while maintaining minimal memory overhead.

## 3   Motivation

Existing KV cache compression methods typically operate at the *token-level*, where tokens considered less important—often determined by heuristics such as attention scores—are evicted during the prefilling stage. However, recent studies suggest that the relevance of prefilled tokens is not static throughout the decoding process Tang et al. (2024). Evicting tokens prematurely based on their initial perceived importance can degrade inference accuracy since tokens initially deemed irrelevant might later become crucial during decoding. To address this dynamic nature, recent studies propose retrieving essential prefilled tokens dynamically based on their relevance to the current decoding query. To mitigate computational overhead associated with searching across all prefilled tokens, these methods typically segment the prefilling text into fixed-size chunks, retrieving and utilizing only the most relevant chunks during decoding.

However, these chunk-based strategies lack a principled justification for preserving semantic coherence, as fixed-size chunks often fragment semantic content arbitrarily. Intuitively, sentences encapsulate complete semantic units, making them natural candidates for maintaining context coherence. To empirically validate this intuition, we examined sentences from distinct categories in the Pile dataset Gao et al. (2020) and measured their embedding similarities in LLMs. As illustrated in Figure 1, sentences from different semantic categories exhibit clearly distinguishable embedding patterns. Moreover, during the decoding phase, we observed that the model-generated query tends to focus attention primarily on semantically related sentences rather than arbitrary token segments. This observation motivates our sentence-level caching approach, which segments the prefilling text into semantically coherent sentence units. During decoding,

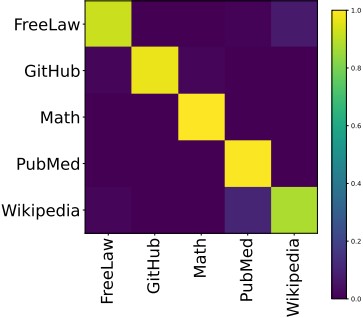

Figure 1: Similarities between sentence embeddings from different categories in the Pile datasets Gao et al. (2020).

we leverage the sentence currently being generated to dynamically identify and recall the most semantically relevant sentence-level cache entries, thereby efficiently preserving context and enhancing inference accuracy.

## 4   SentenceKV

To address the limitations of token-level KV cache compression, we propose **SentenceKV**, which reorganizes CPU cache at the *sentence-level*, leveraging semantic coherence rather than individual tokens. SentenceKV consists of two main phases: **prefilling** and **decoding**. Figure 2 illustrates our framework, with pseudocode provided in Appendix A.2.

### 4.1   Prefilling Stage

We first split the input text into sentences according to punctuation, creating multiple *sentence buckets*, each containing a varying number of tokens. For example, a 32K token input might be split into hundreds of sentence buckets of different lengths. While token importance changes dynamically during decoding, maintaining all tokens at full precision in GPU memory is inefficient and unnecessary. To determine which tokens to retain, we apply an *observation window* Li et al. (2025) for token importance calculation as follows:

**Token importance measurement.**   For an input prompt of length $L$ (e.g., $L = 32K$), we designate the last $N$ tokens (typically $N = 32$) as our observation window. We perform a forward pass where each token in positions $(L - N + 1)$ to $L$ calculates attention scores with respect to all preceding tokens in positions 1 to $(L - N)$. For each token $i$ in the preceding positions, we calculate its importance score $\alpha_i$ by summing the attention scores it receives from all tokens in the observation window, across all the attention heads.

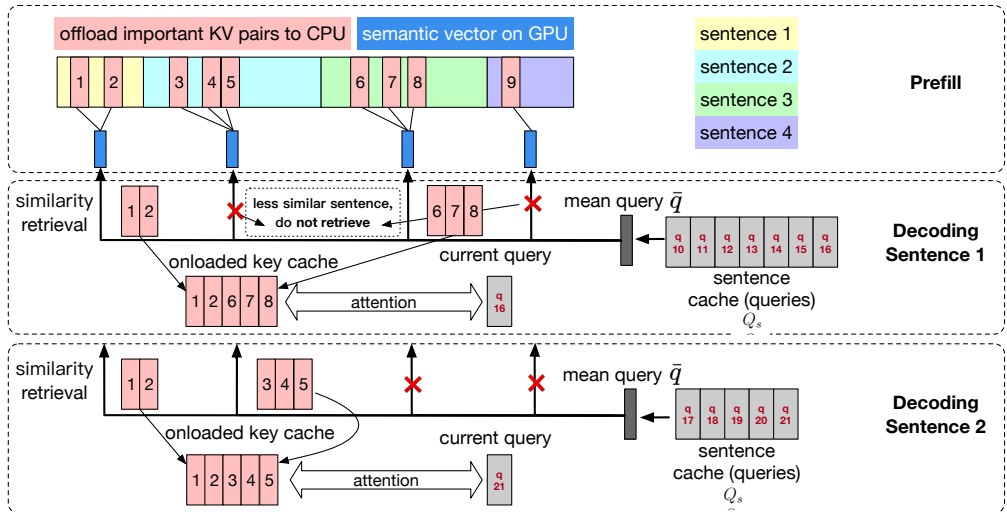

Figure 2: The figure illustrates the two-phase SentenceKV caching mechanism. (1) During the **prefilling phase**, the input prompt is segmented into sentences. Token importance scores are computed, and only the top $\lfloor r \cdot \tau \rfloor$ tokens are offloaded to CPU while the rest are discarded. Semantic vectors representing each sentence are computed and stored. (2) During the **decoding phase**, an empty sentence cache $Q_s$ is initialized. As new tokens are generated, their queries $q_t$ are appended to $Q_s$. At each decoding step, a mean query $\bar{q}$ is computed for the current sentence cache. (3) The similarity between $\bar{q}$ and stored sentence semantic vectors are evaluated, and tokens from the most relevant sentence buckets are retrieved along with their corresponding KV pairs, subject to the token budget $\tau$. (4) Finally, attention outputs are computed, and $Q_s$ is reset when the generated sentence ends.

**Token selection within sentence buckets.** We set a token budget $\tau$ (e.g., $\tau = 1024$) that represents the maximum number of tokens we can keep in GPU memory during decoding. Since tokens that appear unimportant during prefilling may become relevant at later decoding steps, we introduce the *semantic keeping factor r* (typically $r = 2$ or 3), which controls how many tokens are initially retained before selective pruning during decoding. By setting $r > 1$, we retain more tokens than our *final budget ($\tau$)* allows, giving us a richer semantic pool to select from during decoding. We select the top $\lfloor r \cdot \tau \rfloor$ tokens with the highest $\alpha_i$ values across all sentence buckets. This design ensures that the method works when sentence lengths are unbalanced; see Appendix A.7 for the analysis.

**Sentence-level semantic representation.** For each sentence bucket $s$, after identifying its important tokens, we compute a mean key vector for each attention head $h$:

$$\bar{k}_{s,h} = \frac{1}{|S_s|} \sum_{x \in S_s} k_{x,h} \tag{1}$$

where $S_s$ contains the indices of retained tokens in sentence $s$, and $k_{x,h}$ is the key vector of token $x$ in head $h$. These compact sentence-level semantic vectors remain on the GPU for efficient similarity calculations during decoding.

**Memory management.** Rather than permanently discarding tokens, we keep all $\lfloor r \cdot \tau \rfloor$ selected tokens along with their KV pairs, but offload them to CPU memory. During decoding, we will selectively retrieve only the most relevant sentence buckets, up to the token budget $\tau$, based on semantic similarity to the current generation context.

This approach maintains sentence-level semantic coherence while efficiently managing memory. For instance, in a 32K context with $\tau = 1024$ and $r = 2$, we might initially retain 2048 tokens distributed across sentence buckets, but only load the most relevant subset (up to 1024 tokens) to GPU during each decoding step.

## 4.2 Decoding Stage

During the decoding (generation) phase, SentenceKV employs a dynamic mechanism to retrieve the most semantically relevant information from the prefilled tokens.

**Sentence-level query aggregation.** As the model generates new tokens, we aggregate their query vectors into a temporary *sentence cache* $Q_s$. This cache accumulates query vectors $q_t$ for each newly generated token until a sentence boundary (e.g., period, question mark) is detected. Once a complete sentence has been generated, we compute a mean query vector that represents the semantic center of the generated sentence:

$$\bar{q} = \frac{1}{|Q_s|} \sum_{t \in Q_s} q_t, \tag{2}$$

This aggregation captures the overall semantic intent of the generated sentence rather than relying on individual token queries, which may contain position-specific or token-specific noise.

**Semantic similarity and token retrieval.** Using this sentence-level query representation $\bar{q}$, we compute semantic similarity scores between the generated sentence and each stored sentence bucket from the prefilling stage. For each attention head $h$ and sentence bucket $s$, we calculate $\mathcal{S}(\bar{q}, \bar{k}_{s,h}) = \bar{q}^T \cdot \bar{k}_{s,h}$ where $\bar{k}_{s,h}$ is the mean key vector of sentence $s$ for head $h$, as computed during prefilling (Equation 1). This calculation identifies which sentence buckets from the original prompt are most semantically relevant to the current generation context.

We then rank all sentence buckets by their similarity scores and retrieve tokens from the most relevant buckets in descending order of similarity. For example, if the highest similarity scores are for sentences containing financial data, we prioritize retrieving tokens from those sentence buckets first. We continue this process until we reach our token budget $\tau$, ensuring we load only the most contextually relevant information back to the GPU.

**Attention computation with retrieved tokens.** CPU pairs associated with the retrieved tokens are loaded from CPU back to the GPU memory. For the current query vector $q$, attention is computed using only these retrieved tokens:

$$O = \text{softmax}\left(\frac{q K_\tau^\top}{\sqrt{d}}\right) V_\tau, \tag{3}$$

where $(K_\tau, V_\tau)$ are the key and value matrices for the retrieved tokens (up to $\tau$ tokens), and $d$ is the head dimension. This focused attention computation significantly reduces computational overhead by considering only the most relevant context. After generating the next token, we append its query to $Q_s$ and repeat the process. When a new sentence boundary (e.g., period, question mark) is detected, we reset $Q_s$ and begin accumulating queries for the next sentence. Compared to existing compression strategies, SentenceKV preserves richer semantic information by treating entire sentences as retrieval units, mitigating fragmentation, and allowing dynamic reactivation of offloaded tokens when needed.

## 5 Experiments

In this section, we present the experimental setup and evaluate our proposed method, **SentenceKV**, on several benchmarks. We compare it with state-of-the-art KV cache compression methods using metrics such as accuracy, perplexity (PPL), memory usage, and latency.

All the experiments are conducted using two NVIDIA H100 80GB GPUs and two NVIDIA H100 NVL 96GB GPUs. We evaluate our method on four benchmarks: **PG-19** (language modeling task) Rae et al. (2019), **LongBench** Bai et al. (2023), **NIAH** (retrieval of a hidden "needle") Kamradt (2023), **RULER** Hsieh et al. (2024). Our experiments use the models **Llama-3-8B** AI (2024a), **Llama-3.1-8B-Instruct** AI (2023), **Llama-3.2-3B-Instruct** AI (2024b) and **Longchat-v1.5-7B** LMSYS (2024). We compare **SentenceKV** with several baselines,

Table 1: Performance comparison between SentenceKV and baseline methods on LongBench tasks. Token budget is set to 1024; '*' indicates out-of-memory (OOM) errors, **bold** values are the maximum among non-Full KV methods.

| | LLMs | Single-Document QA | | | Multi-Document QA | | | Few-shot Learning | | | Synthetic | | Code | |
| | | NrtvQA | Qasper | MF-en | HotpotQA | 2WikiMQA | Musique | TREC | TriviaQA | SAMSum | PCount | PRe | Lcc | RB-P |
|---|---|---|---|---|---|---|---|---|---|---|---|---|---|---|
| **Llama-3.1** | Full KV | 29.59 | 47.52 | 53 | 53.76 | 46.12 | 28.38 | 7.5 | 89.41 | 7.47 | 6.25 | 99.5 | 13.52 | 11.72 |
| | H2O | * | * | 47.89 | * | * | * | 12.5 | * | * | * | 98 | * | * |
| | SnapKV | 27.2 | 46.28 | 52.41 | 52.66 | 45.67 | **28.63** | 6.5 | 89.41 | 7.51 | 4.42 | **99.5** | 13.75 | 11.93 |
| | Quest | 22.49 | 45.2 | 48.72 | 51.94 | 44.02 | 26.78 | **13.5** | 88.63 | **10.31** | 4.21 | 97 | **14.79** | 16.64 |
| | InfLLM | 27.64 | 45.32 | 50.6 | 51.47 | 44.59 | 23 | 12.5 | **90.92** | 6.79 | 3.5 | 95 | 13.33 | **23.79** |
| | SentenceKV | **29.53** | **47.49** | **53.15** | **53.39** | **45.8** | 28.04 | 11.5 | 89.41 | 7.48 | **5.28** | **99.5** | 13.44 | 11.28 |
| **Longchat-v1.5** | Full KV | 22.68 | 33.99 | 46.32 | 39.22 | 26.95 | 14.57 | 85.71 | 84.93 | 22.34 | 0 | 18.5 | 52.94 | 56.89 |
| | H2O | * | * | * | * | * | * | * | * | * | * | * | * | * |
| | SnapKV | 20.79 | 30.23 | 41.98 | 36.41 | **26.81** | 13.38 | 64.5 | 80.88 | **26.79** | 0 | 17.5 | 52.02 | 55.41 |
| | Quest | 20.76 | 31.67 | 41.55 | 37.25 | 25.47 | **13.65** | 64 | **81.83** | 25.73 | **3.5** | 17 | 49.1 | 51.2 |
| | InfLLM | 13.59 | 24.87 | 35.87 | 26.52 | 21.33 | 9.42 | 49.0 | 75.56 | 21.08 | 0 | 10.83 | 27.62 | 16.09 |
| | SentenceKV | **22.01** | **32.98** | **43.29** | **37.49** | 26.17 | 13.29 | **64.5** | 76.54 | 24.29 | 0 | 16.5 | **54.33** | **56.23** |
| **Llama-3.2-3B** | Full KV | 23.53 | 41.69 | 49.82 | 49.96 | 41.8 | 20.96 | 2.0 | 9.17 | 5.78 | 7.5 | 78.5 | 26.31 | 25.2 |
| | H2O | * | * | * | * | * | * | * | * | * | * | * | * | * |
| | SnapKV | 20.02 | 35.17 | 48.29 | **48.98** | 37.63 | 20.12 | 0.5 | 9.07 | 6.19 | 4.35 | 81.5 | 26.19 | 25.36 |
| | Quest | 15.53 | 31.7 | 41.36 | 46.07 | 33.25 | 19.12 | 3.5 | 13.17 | **8.36** | **5.17** | 67.0 | **28.64** | **28.45** |
| | InfLLM | 20.48 | 33.97 | 45.89 | 45.94 | 36.7 | 16.54 | 0.0 | **48.77** | 3.56 | 2.0 | 36.5 | 24.86 | 26.27 |
| | SentenceKV | **24.12** | **38.38** | **49.38** | **48.98** | **40.01** | **20.95** | **6.0** | 8.82 | 6.06 | 0.0 | **84.0** | 26.64 | 23.46 |

including **H2O** Zhang et al. (2023), **SnapKV** Li et al. (2025), **Quest** Tang et al. (2024), **InfLLM** Xiao et al. (2024), **ShadowKV** Sun et al. (2024), as well as a **Full KV cache** baseline. The evaluation metrics include model accuracy, GPU memory usage (in GB), and latency (in ms/token).

## 5.1 Language Modeling

**Setup.** We evaluate the PG-19 dataset to measure PPL under context lengths of up to 32k tokens. A lower PPL indicates that the model is less surprised by ground truth tokens, reflecting higher confidence in its predictions. In our setup, the model is fed sequences exceeding 32k tokens from the dataset, followed by autoregressive decoding, during which PPL is computed at each step. Lower PPL values correspond to better predictive performance.

**Results.** Figure 3 shows PPL measured at a token budget of $\tau = 1024$ (3% of the full KV cache when the context length is 32k). Despite this significant compression, SentenceKV consistently achieves PPL nearly identical to the full KV cache baseline, even as the input length grows. This demonstrates that SentenceKV effectively retains essential contextual information, preserving generation quality while dramatically reducing GPU memory usage. The minimal performance gap across all context lengths highlights the strength of semantic-level caching in maintaining decoding fidelity under tight memory constraints.

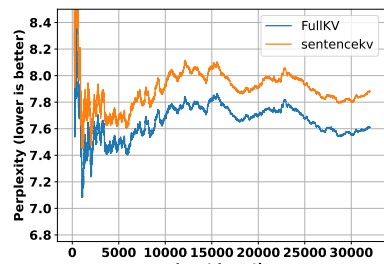

Figure 3: PPL on PG-19 with a token budget $\tau = 1024$. Lower PPL indicates better performance.

## 5.2 LongBench

**Setup.** We select a suite of tasks from the LongBench benchmark, including single-document QA, multi-document QA, and few-shot learning, as well as synthetic and code generation tasks. Each task features input sequences ranging from a few thousand to tens of thousands of tokens.

**Results.** Table 1 reports accuracy on LongBench tasks. The full KV cache achieves the best performance but is impractical for long inputs due to high GPU memory usage. In contrast,

**SentenceKV** achieves comparable accuracy across most tasks while operating under a strict token budget, demonstrating strong performance-memory trade-offs. It consistently matches or closely approaches full KV cache on both Llama models and LongChat-v1.5-7B. Compared to other baselines, SentenceKV is more stable: H2O frequently fails with OOM errors on long inputs (marked as * in the table). SnapKV and InfLLMperforms well but slightly lags behind SentenceKV on several tasks. Quest does not perform well on GQA-based models (including Llama-3.1). Overall, SentenceKV provides an effective solution for long-context inference with reduced memory usage.

## 5.3 Needle In A Haystack

**Setup.** In the NIAH benchmark, a single critical sentence (the "needle") is placed within a large, mostly irrelevant context. The task requires the model to retrieve this key sentence and answer a related question. We evaluate the method using the retrieval accuracy (%) metric, which indicates whether SentenceKV correctly identifies the crucial sentence.

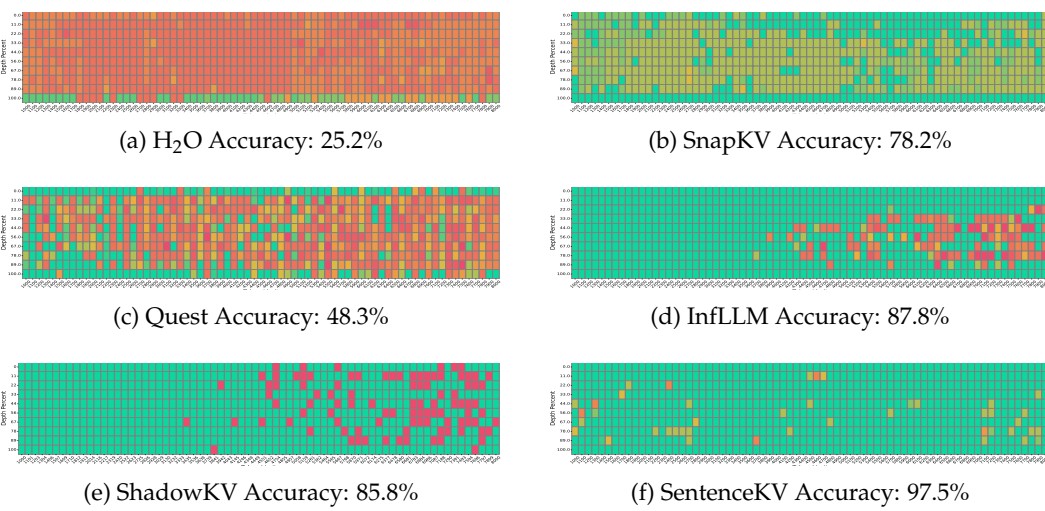

(a) H$_2$O Accuracy: 25.2%

(b) SnapKV Accuracy: 78.2%

(c) Quest Accuracy: 48.3%

(d) InfLLM Accuracy: 87.8%

(e) ShadowKV Accuracy: 85.8%

(f) SentenceKV Accuracy: 97.5%

Figure 4: Retrieval accuracy on the NIAH benchmark across context lengths up to 8000 tokens, with a token budget $\tau = 128$.

**Results.** Figure 4 shows the retrieval accuracy results on the NIAH benchmark, comparing SentenceKV against H2O, SnapKV, Quest, InfLLMand ShadowKVwith a token budget $\tau = 128$ and context lengths up to 8000 tokens, with Llama-3-8B model. SentenceKV achieves an outstanding accuracy of 97.5%, significantly outperforming other methods. SnapKV, InfLLM and ShadowKV shows reasonable accuracy (78.2%, 87.8% and 85.8%), whereas Quest provides moderate performance (48.3%). The reason for Quest's low accuracy is analyzed in Appendix A.4. Notably, H2O exhibits poor performance (25.2%), primarily due to its limitations in handling extensive contexts, further highlighting the advantage of SentenceKV in accurately and efficiently retrieving critical sentences within lengthy texts.

## 5.4 RULER

**Setup.** Following the LongBench and NIAH evaluations, we additionally benchmark SentenceKV on the RULER long-context retrieval suite. Each example contains up to 64K tokens, pushing GPU memory pressure well beyond the typical setting. We use Llama-3.1-8B-Instruct with flash–attention enabled and compare against representative KV-cache baselines.

**Results.** Table 2 reports retrieval accuracy (higher is better) on eight RULER subtasks. SentenceKV matches or exceeds every offloading baseline and trails the full kv cache upper-bound on specific tasks. ShadowKV attains comparable accuracy, but incurs longer prefill

time owing to its per-sentence SVD reconstruction; in contrast, SentenceKV keeps the pipeline simple, light-weight, and easier to integrate.

Although a fixed semantic keeping factor already performs well, we notice task-dependent variation. An adaptively chosen semantic keeping factor may increase the accuracy; we leave a full exploration of this adaptive strategy to future work. Overall, these RULER results confirm the scalability and robustness of SentenceKV in extreme long-

| Method | S1 | S2 | MK1 | MQ | MV | FWE | VT | QA1 |
|---|---|---|---|---|---|---|---|---|
| FullKV | 1.00 | 1.00 | 1.00 | 0.98 | 0.96 | 0.85 | 0.94 | 0.55 |
| SnapKV | 1.00 | 0.94 | 0.96 | 0.82 | 0.76 | 0.55 | 0.82 | 0.63 |
| H$_2$O | OOM | OOM | OOM | OOM | OOM | OOM | OOM | OOM |
| Quest | 1.00 | 0.98 | 1.00 | 0.82 | 0.87 | 0.61 | 0.74 | 0.56 |
| InfLLM | 1.00 | 0.20 | 0.14 | 0.17 | 0.22 | 0.81 | 0.43 | 0.12 |
| ShadowKV | 1.00 | 0.98 | 1.00 | 0.87 | 0.74 | 0.78 | 0.79 | 0.70 |
| SentenceKV | 1.00 | 0.98 | 0.98 | 0.77 | 0.78 | 0.61 | 0.86 | 0.63 |

Table 2: Retrieval accuracy on the **RULER** benchmark (64K tokens). "OOM" indicates out-of-memory on a single H100 80 GB GPU.

context scenarios and motivate its deployment in real-world applications where very long contexts are becoming the norm.

## 5.5 Efficiency

We evaluate SentenceKV against Full KV cache, SnapKV, and SqueezeKV, focusing on GPU memory usage and inference latency (Figure 5). SnapKV performs KV eviction only during the prefill phase without dynamic retrieval during the decoding, resulting in minimal latency. Therefore, we focus our end-to-end comparison primarily with Full KV and SnapKV.

Using the Llama-3.1-8B-Instruct model, SentenceKV achieves substantial memory savings as context length increases. At 256k tokens, it uses 52.71 GB of memory, compared to 89.71 GB for Full KV. Its inference latency remains stable at around 17.8 ms, significantly lower than Full KV's 84.9 ms. Overall, SentenceKV supports longer contexts under constrained GPU memory while maintaining low latency, making it a practical and efficient solution for long-context decoding. See Appendix A.3 for results of the LongChat-v1.5-7B model. A detailed efficiency and accuracy tradeoff analysis is shown in Appendix A.5, and a detailed GPU profiling is shown in Appendix A.6.

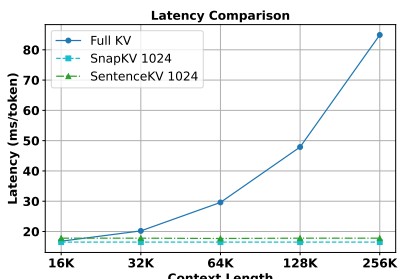

(a) Latency Comparison

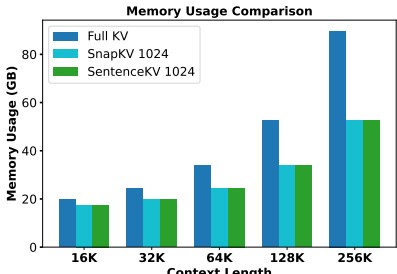

(b) Memory Usage Comparison

Figure 5: Efficiency comparison of KV caching methods ($\tau = 1024$).

## 6 Ablation Studies

To further validate the design choices behind SentenceKV, we perform ablation studies focusing on two key components: sentence chunking and the cache strategy.

## 6.1 Sentence chunking

An alternative strategy to our sentence-level chunking is to perform equal-sized chunking based on the total number of sentences, dividing the text uniformly. However, this approach consistently yields worse results on LongBench (see Table 3), likely due to the loss of coherence and topical alignment within chunks. When sentences are split arbitrarily, important semantic relationships and narrative flow can be disrupted, making it harder for the model to maintain context and generate meaningful outputs.

In contrast, our approach leverages semantic segmentation to group related information together, preserving logical boundaries and thematic continuity. This experiment demonstrates that chunking based on semantic structure leads to better contextual integrity and improved downstream performance.

Table 3: Equal-size chunks perform worse than SentenceKV on LongBench tasks.

| Task | Equal Chunks | SentenceKV |
|------|--------------|------------|
| NrtvQA | 27.02 | 29.59 |
| HotpotQA | 53.19 | 53.76 |
| TREC | 8.5 | 11.5 |
| PCount | 4.81 | 5.28 |

## 6.2 Semantic keeping factor and cache strategy

Figure 6 illustrates how the semantic keeping factor—which we defined as the ratio $r$ between the number of tokens retained per sentence during the prefilling stage and the fixed decoding token budget—influences retrieval accuracy on the NIAH benchmark. As the semantic keeping factor increases, the model retains a broader context, leading to improved accuracy—up to a point. However, overly high values may introduce redundant or noisy information, highlighting the importance of careful parameter tuning to balance recall and precision.

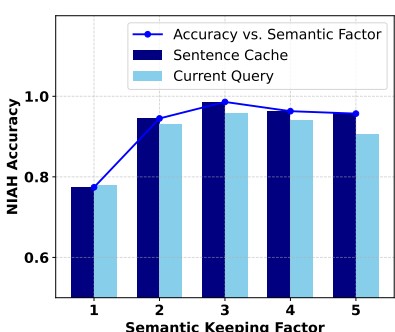

Figure 6: Effect of semantic keeping factor and caching strategy.

The figure also compares two retrieval strategies: using the current token's query versus the mean query aggregated from the sentence cache. The latter consistently outperforms the former, demonstrating that sentence-level aggregation yields a more stable and semantically representative signal for selecting relevant sentence buckets.

## 7 Conclusion

In this work, we introduced **SentenceKV**, a sentence-level semantic KV caching framework that improves inference efficiency for large language models on long contexts. SentenceKV integrates sentence-level processing into token-level transformer computations by grouping tokens into coherent sentences. It compresses the KV cache into compact sentence embeddings on GPU and offloads full KV pairs to CPU. During decoding, only the most relevant sentences are retrieved within a fixed token budget, preserving coherence and reducing data transfer. Evaluations on PG19, LongBench, NIAH and RULER show that SentenceKV matches full-cache performance in perplexity and task accuracy using as little as 3% of the cache, while reducing GPU memory usage by over 30% and maintaining low latency up to 256K tokens. In retrieval-heavy settings, it improves retrieval accuracy by over 20% compared to prior methods, effectively identifying critical information in long inputs.

**Future Work.** Future work could explore several promising directions. First, relying on punctuation for sentence segmentation can lead to inaccurate boundaries, especially in texts with sparse or irregular punctuation, which harms semantic coherence. Using NLP-based segmentation tools, such as those in the Natural Language Toolkit Bird (2006), could improve accuracy. Second, the fixed semantic factor may reduce adaptability to texts with varying sentence lengths and structures. Third, many recent reasoning models generate chain-of-thought (CoT) traces with each sentence representing a self-contained logical step, making SentenceKV a natural fit. Future evaluations could target dedicated CoT datasets such as DeepSeek-R1 Guo et al. (2025) and GSM-Infinite Zhou et al. (2025) to assess this potential. Lastly, incorporating lightweight on-device similarity search or mixed-precision sentence embeddings could reduce CPU–GPU round-trip overhead under long-context scenarios.

## Acknowledgments

We thank the anonymous reviewers for their valuable suggestions, particularly on evaluating additional benchmarks, as well as their constructive comments on analyzing the efficiency of our method. We also thank Zhenyu Liu from the Department of Electrical, Computer, and Systems Engineering at Rensselaer Polytechnic Institute for his guidance on the implementation details.

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

# A Appendix

## A.1 Memory Usage Calculation at Decoding Stage

In modern transformer-based large language models (LLMs), each input token typically produces a key ($K$) and value ($V$) vector for every attention head in each layer. As the context length grows, the dimension of the KV cache stored in the GPU memory also grows linearly. Formally, let $L$ be the initial prompt length, $M$ the number of Transformer layers, and $H$ the number of attention heads, each with dimension $d$. At each decoding step $t$, the GPU memory cost, denoted by $\mathrm{Cost}(t)$, increases as follows:

$$\mathrm{Cost}(t) = O(M \times H \times (L + t) \times d), \tag{4}$$

where $(L + t)$ indicates the total number of processed tokens ($L$ tokens in the prompt plus $t$ tokens generated). When $L$ is very large (e.g., thousands of tokens), this memory cost can quickly exceed the available GPU capacity, leading to performance degradation or even inference failure.

## A.2 SentenceKV: pseudocode

---

**Algorithm 1** SentenceKV Attention Mechanism

---

**Require:** Prompt tokens, token budget $\tau$, semantic keeping factor $r$, observation window size $N$
1: **Prefilling Phase:**
2: Split prompt into sentences based on punctuation or linguistic cues
3: **for** each sentence $s$ **do**
4:     Compute token importance $\alpha_i$ for tokens $i$ in $s$ using the last $N$ tokens
5:     Retain top $\lfloor r \cdot \tau \rfloor$ tokens based on $\alpha_i$ and discard the rest
6:     Compute mean key vector $\bar{k}_{s,h}$ for each head $h$ (Eq. 1)
7:     Offload a small subset ($r \cdot \tau$) of tokens to CPU for retrieval
8: **end for**
9:
10: **Decoding Phase:**
11: Initialize an empty sentence cache $Q_s$
12: **for** $t = 1, 2, \ldots$ **do**
13:     Generate next token $x_t$ and compute query $q_t$
14:     Append $q_t$ to $Q_s$
15:     Compute $\bar{q}$ for the current sentence (Eq. 2)
16:     Compute similarity $\mathcal{S}(\bar{q}, \bar{k}_{s,h})$ for each sentence bucket
17:     Retrieve top-$\tau$ tokens from most relevant sentence buckets by similarity
18:     Load keys and values of the corresponding $\tau$ tokens to GPU from CPU
19:     $O_t = \mathrm{softmax}\left(\frac{qK_\tau^\top}{\sqrt{d}}\right) V_\tau$ (Eq. 3)
20:     **if** a sentence boundary is reached **then**
21:         Reset $Q_s$ for the next sentence
22:     **end if**
23: **end for**

---

## A.3 Latency and Memory Footprint

For the Longchat-v1.5-7B LMSYS (2024) model, SentenceKV similarly demonstrates considerable efficiency gains. The peak memory footprint at the maximum context length (256k) is reduced from the OOM (out-of-memory) scenario encountered by Full KV caching to a manageable 48.31 GB. Latency also remains stable around 16.9 ms, whereas Full KV caching becomes infeasible beyond 64k tokens due to excessive memory demands. The details of the results are shown in the Appendix Table 4 and Table 5.

Table 4: Latency and memory usage. Results are from the Llama-3.1-8B-Instruct model with a token budget of $\tau = 1024$.

| Latency (ms/token) | | | | | |
| --- | --- | --- | --- | --- | --- |
| **Method** | **16K** | **32K** | **64K** | **128K** | **256K** |
| Full KV | 16.8ms | 20.2ms | 29.6ms | 47.9ms | 84.9ms |
| SnapKV 1024 | 16.5ms | 16.5ms | 16.5ms | 16.5ms | 16.5ms |
| SentenceKV 1024 | 17.8ms | 17.8ms | 17.7ms | 17.8ms | 17.8ms |
| Memory Usage (GB) | | | | | |
| **Method** | **16K** | **32K** | **64K** | **128K** | **256K** |
| Full KV | 19.78GB | 24.45GB | 33.77GB | 52.42GB | 89.71GB |
| SnapKV 1024 | 17.47GB | 19.82GB | 24.52GB | 33.92GB | 52.71GB |
| SentenceKV 1024 | 17.47GB | 19.83GB | 24.54GB | 33.92GB | 52.71GB |

Table 5: Latency and memory usage. Results are from the Longchat-v1.5-7B model with a token budget of $\tau = 1024$.

| Latency (ms/token) | | | | | |
| --- | --- | --- | --- | --- | --- |
| **Method** | **16K** | **32K** | **64K** | **128K** | **256K** |
| Full KV | 29.2ms | 47.3ms | 84.2ms | OOM | OOM |
| SnapKV 1024 | 16.5ms | 16.5ms | 16.5ms | 16.5ms | 16.5ms |
| SentenceKV 1024 | 16.6ms | 16.6ms | 16.5ms | 16.8ms | 16.9ms |
| Memory Usage (GB) | | | | | |
| **Method** | **16K** | **32K** | **64K** | **128K** | **256K** |
| Full KV | 27.96GB | 43.26GB | 73.88GB | OOM | OOM |
| SnapKV 1024 | 15.18GB | 17.73GB | 21.52GB | 30.45GB | 48.30GB |
| SentenceKV 1024 | 15.18GB | 17.73GB | 21.53GB | 30.45GB | 48.31GB |

## A.4 Quest Sensitivity to Chunk Size

The **Quest** Tang et al. (2024) cache-offloading baseline divides the context into fixed-length chunks and retains only a subset per chunk. Prior work typically uses a chunk size of 8 tokens, but this granularity differs markedly from SentenceKV, where one bucket naturally corresponds to a full sentence ($\sim 25$–30 tokens on our corpora). To ensure a fair comparison on NIAH Kamradt (2023), we therefore re-evaluated Quest with larger chunk sizes that better match the average sentence length.

| Method | Quest (Chunk Size 32) | Quest (Chunk Size 16) | SentenceKV (Adaptive) |
| --- | --- | --- | --- |
| NIAH Acc. % | 48.3 | 96.1 | 97.5 |

Table 6: Quest results on NIAH under different settings. Model: Llama-3.1-8B with a token budget $\tau = 128$.

Quest proves highly sensitive to chunk granularity: reducing the chunk size from 32 to 16 tokens nearly doubles accuracy (Table 6). In contrast, **SentenceKV** employs adaptive buckets (median length 25–30 tokens) and attains 97.5 % accuracy with no manual tuning, highlighting the robustness of sentence-aligned grouping.

## A.5 Efficiency/Accuracy Trade Off

Profiling a 64 k-token `niah_single` input on an H100-96 GB NVL shows: prefill bucket construction costs 4 s once; each decoding step adds 0.019 s (GPU→CPU 2048 tokens), 0.0038 s (CPU→GPU 1024 tokens), and 0.050 s for similarity ranking, totalling 0.073 s. These

overheads are negligible relative to the $> 60$ ms gap between SentenceKV and Full KV, confirming a favourable accuracy–latency–memory balance for real-world long-context serving.

Fixing $\tau{=}128$ and semantic-keeping factor $r{=}3$ on NIAH, SnapKV decodes in 15 ms per token; SentenceKV needs only $+1$ ms extra yet boosts accuracy by 25% (Table 7). Such minor latency growth is often preferable to the sharp accuracy drop that accompanies more aggressive compression.

|  | SnapKV | SentenceKV | Increase |
|---|---|---|---|
| Latency (ms/token) | 15 | 16 | 6% |
| NIAH Acc. (%) | 78.2 | 97.5 | +25% |

Table 7: Trade-off on NIAH ($\tau = 128$, $r = 3$).

### A.6 GPU Profiling Details

To illustrate how SentenceKV achieves reductions in both memory *and* latency, we present a fine-grained GPU profile showing (i) the set of KV cache blocks retained in GPU memory during decoding, and (ii) the breakdown of decoding time at a 64K-token context length.

**Memory composition at 256 K context (Llama-3.1-8B).** Peak usage reported by `torch.cuda.max_memory_allocated()` drops from 89.71 GB (Full KV) to 52.71 GB with SentenceKV. The residual footprint consists of:

(1) *KV cache (subset only):* 33.55 GB $\rightarrow$ 0.10 GB after retrieving just $\sim$1K top tokens per head.

(2) *Sentence-level semantic vectors:* tens of MB; negligible relative to the model.

(3) *Temporary tensors:* short-lived attention buffers, now smaller because attention is computed on far fewer keys/values.

(4) *Model weights:* feed-forward and projection matrices dominate the remaining 50+ GB; they stay on-GPU for all methods.

The reported peak includes allocator fragmentation and transient buffers, yet the *relative* 37 GB reduction clearly evidences SentenceKV's compression efficacy.

**Latency breakdown at 64 K context (RULER `niah_single`).** Profiling on an H100-96 GB NVL with $\tau = 1024$, $\kappa = 2$ yields:

(1) **Prefill bucket construction** (one-time): 4 s for sentence grouping and top-$k$ token selection.

(2) **CPU$\leftrightarrow$GPU transfer** (per step): 0.0191 s offload 2048 tokens + 0.0038 s onload 1024=0.0229 s.

(3) **Sentence ranking** (per step): 0.050 s (CPU dense similarity search).

Even if the 0.073 s per-step overhead is conservatively added to the computation latency shown in Figure 5, SentenceKV remains markedly faster than Full KV because drastically fewer keys and values participate in attention.

In summary, the large KV-cache shrink (33.55 GB$\rightarrow$0.10 GB) alleviates the memory bottleneck in long-context attention, simultaneously cutting decoding time while staying within modest GPU capacity.

### A.7 Effect of Sentence Length

SentenceKV ranks sentences by relevance until the remaining token budget $\tau$ is exhausted. One possible drawback is that an unusually long but highly relevant sentence could consume a disproportionate share of the budget, thereby excluding other useful context.

To address this, we apply SnapKV-style Li et al. (2025) prefiltering globally across all input tokens during the prefill stage. Specifically, instead of selecting top tokens from each sentence individually, SnapKV selects the top $r \cdot \tau$ most important tokens from the entire

document, regardless of sentence boundaries. As a result, the number of retained tokens per sentence is determined by its relative semantic contribution, rather than its raw length. Even if a sentence is long, it will retain fewer tokens if it is not highly informative.

In practice, this global filtering strategy significantly reduces token retention from long but unimportant sentences. Our empirical analysis of LongBench shows that sentence lengths are generally concentrated around the mean, with very long sentences being rare. After filtering, token counts per sentence are typically modest and broadly distributed, helping to avoid scenarios where a single sentence dominates the token budget.

Future work could include detecting abnormally long retained token spans ("buckets") during prefill and applying simple mitigation strategies. For example, one could compute the mean and standard deviation of sentence lengths in the input, and if a sentence exceeds mean $+ n \times$ std, split it into smaller sub-spans. Such a lightweight check would add minimal computational overhead while preventing outlier buckets from skewing the retrieval process.

