# OpenReview forum: "SentenceKV: Efficient LLM Inference via Sentence-Level Semantic KV Caching"
_colmweb.org/COLM/2025/Conference — COLM 2025_

### Official Review · Reviewer_ecqc · 2025-04-11

**Rating:** 5
**Confidence:** 4
**Ethics Flag:** 1

**Summary:**

This paper proposes SentenceKV, a sentence-level semantic caching method to improve long-context LLM inference efficiency.

SentenceKV groups tokens into sentences during the prefilling stage, computes mean key vectors for each sentence, and keeps these semantic vectors on GPU. During decoding, it retrieves relevant sentence KV pairs from CPU based on the similarity between the current generation's aggregated query vector and the stored sentence vectors.

While the idea of leveraging sentence-level semantics is intuitive and potentially valuable, I found several critical issues in both the system evaluation and the correctness of the experimental setup that, in my opinion, make the current version not ready for publication.

**Reasons To Accept:**

- The core insight that sentences are natural semantic units is sound. Compared to fixed-size token chunks, sentence-based grouping can potentially lead to more coherent context retrieval.
- The method is relatively simple to implement and avoids heavyweight techniques like clustering or learned memory routing. The use of mean query vectors and semantic similarity scoring makes the approach interpretable.
- Even if some aspects need refinement, the idea of abstracting KV cache to higher-level units could spark further research in retrieval-based inference.

**Reasons To Reject:**

- The reported latency reduction is surprisingly large. In Figure 5, SentenceKV reportedly brings down decoding latency from 84.9ms to 17.8ms at 256K tokens, while GPU memory usage goes from 89.71GB to 52.71GB. This does not align with our understanding of decoding in LLMs (memory bound). Memory usage is reduced by ~40%, but latency is reduced by almost 5x?  This undercuts the claimed efficiency gains. A deeper breakdown (e.g., profiling results, CPU-GPU communication time, kernel latency) is needed to make these results convincing.
- Figure 4 shows that Quest performs poorly on the NIAH benchmark with only 48.3% accuracy, while SentenceKV hits 97.5%. This is very surprising. In prior work such as [1], Quest performs reasonably well on needle-style retrieval tasks (RULER Single Needle Retrieval). It is a bit unclear that what setup was used here?
- To validate the effectiveness of retrieval-based caching, a single benchmark like NIAH isn't enough—especially when Quest is performing unexpectedly poorly. I strongly recommend including RULER or similar retrieval-intensive tasks that are better established in the literature.
- In Line 184, the paper mentions selecting top-$r\times \tau$ tokens by attention importance. I am a bit confused: Are these head-specific selections (i.e., each attention head has its own important tokens) or is the importance aggregated across KV head groups or is it shared for all heads?
- The sentence bucket mechanism is supposed to save memory by storing only compact sentence-level vectors on GPU and offloading full KV to CPU. However, in Figure 5, the memory savings from FullKV to SentenceKV is from 89.71GB to 52.71GB. So what exactly remains on GPU during decoding? Are the semantic vectors that large?

[1] ShadowKV: KV Cache in Shadows for High-Throughput Long-Context LLM Inference

---

> ### Author Response · Authors · 2025-06-03
> **Q1. Quest performs poorly on the NIAH.**
>
> We thank the reviewer for pointing this out and agree that clarification is needed.
>
> We found that **Quest's performance is highly sensitive to the choice of chunk size and cache budget**. In prior work, Quest typically uses a chunk size of 8. However, to ensure a fair comparison with SentenceKV—where one sentence typically corresponds to ~17–20 words or ~25–30 tokens—we increased Quest’s chunk size to 32 to align with our sentence-based budget.
>
> To better isolate the impact of chunk size, we also ran experiments with **chunk size = 16**, and observed improved accuracy for Quest on NIAH, though still lower than SentenceKV. This further suggests that Quest is sensitive to chunk granularity, while **SentenceKV dynamically adapts to natural sentence boundaries**, yielding more robust performance across datasets.
>
> | Method                                      | quest (chunk size = 32) | quest (chunk size = 16) | sentencekv (adaptive chunk size, 25–30 on average) |
> |---------------------------------------------|--------------------------|--------------------------|----------------------------------------------------|
> | NIAH Accuracy (%)                           | 48.3                     | 96.1                     | 97.5                                               |

---

> > ### Author Response · Authors · 2025-06-03
> > **Q2. Add retrieval-intensive tasks like the RULER.**
> >
> > We appreciate the reviewer’s suggestion. In response, we have added experiments on **RULER**, a retrieval-intensive benchmark widely used in prior work. These results are now included in the following table and provide a complementary evaluation to NIAH.
> >
> > To ensure a fair comparison, we also adjusted **Quest’s chunk size** appropriately for RULER. Unlike NIAH, where we used a chunk size of 32 to match the typical sentence length, RULER contains much shorter sentences on average. Therefore, we set Quest’s chunk size to **16** on RULER, which aligns better with the sentence-based granularity used by SentenceKV.
> >
> > Besides, we also add another offloading based method - ShadowKV, for a more comprehensive comparison.  ShadowKV is a high-throughput long-context LLM inference system that compresses the key cache using low-rank decomposition (SVD) and offloads the value cache to the CPU. It reconstructs sparse KV representations on the fly to reduce memory usage and support larger batch sizes. However, this comes at the cost of increased implementation complexity and prefill overhead.
> >
> > Below table shows the RULER results with 64k input length.
> >
> > | Method     | S1  | S2  | MK1 | MQ  | MV  | FWE | VT  | QA1 |
> > |------------|-----|-----|-----|-----|-----|-----|-----|-----|
> > | FullKV     | 1   | 1   | 1   | 0.98 | 0.96 | 0.85 | 0.94 | 0.55 |
> > | SnapKV     | 1   | 0.94 | 0.96 | 0.82 | 0.76 | 0.55 | 0.82 | 0.63 |
> > | H2O        | OOM | OOM | OOM | OOM | OOM | OOM | OOM | OOM |
> > | Quest      | 1   | 0.98 | 1   | 0.82 | 0.87 | 0.61 | 0.74 | 0.56 |
> > | ShadowKV   | 1   | 0.98 | 1   | 0.87 | 0.74 | 0.78 | 0.79 | 0.70 |
> > | SentenceKV | 1   | 0.98 | 0.98 | 0.77 | 0.78 | 0.61 | 0.86 | 0.63 |
> >
> > **SentenceKV** achieves comparable or better performance on multiple tasks with a much simpler and faster prefill process. We also note that **SentenceKV and ShadowKV often achieve similar levels of accuracy**, which we consider acceptable and expected given the different design philosophies. ShadowKV performs more fine-grained reconstruction via low-rank SVD and sparse selection, which allows slightly more precise prefill handling—but at the cost of increased prefill time, code complexity, and integration difficulty. In contrast, SentenceKV achieves comparable accuracy while offering a **simpler, faster, and more practical implementation**, which we believe is more suitable for real-world deployment.
> >
> > That said, we also observed some performance variation across tasks. In particular, the **fixed semantic keeping factor** used in our current setup may not always achieve the optimal trade-off between memory usage and retrieval quality.
> >
> > As part of our ongoing and future work, we are actively investigating the relationship between input characteristics—such as **sentence length distribution** and **semantic diversity**—and the optimal semantic keeping factor. Preliminary observations suggest that **dynamically adjusting** this factor based on input statistics can further improve both efficiency and accuracy.
> >
> > We believe this addition strengthens our evaluation and confirms the effectiveness of SentenceKV in retrieval-heavy settings.

---

> > ### Author Response · Authors · 2025-06-03
> > **Q3. The memory savings from FullKV to SentenceKV is from 89.71GB to 52.71GB. What exactly remains on GPU during decoding?**
> >
> > We thank the reviewer for this question. We clarify below what contributes to GPU memory usage during decoding with **SentenceKV**:
> >
> > - **KV cache (subset only):** In Full KV, decoding with a 256K-token context consumes ~33.55 GB for the key and value tensors (per layer × per head). In SentenceKV, we only retrieve and store ~1K top tokens per attention head from CPU to GPU, resulting in a ~0.1 GB KV cache size.
> >
> > - **Sentence-level semantic vectors:** These vectors are relatively small in size and contribute minimally to overall memory usage.
> >
> > - **Temporary tensors:** Additional buffers used during attention computation (e.g., attention weights, masks) are short-lived and small in size due to the compressed KV.
> >
> > - **Model parameters:** The largest contributor to GPU memory is the model itself. For 7B/8B models, feed-forward layers and projection weights are persistently stored on GPU for inference.
> >
> >
> > Finally, we note that the reported GPU memory usage reflects peak memory, as measured by PyTorch’s torch.cuda.max_memory_allocated(). This includes allocator fragmentation and temporary buffer allocations, and therefore may exceed the theoretical sum of all tensor sizes. The key point, however, is the relative reduction in memory usage, which clearly demonstrates the effectiveness of SentenceKV’s compression strategy.

---

> > ### Author Response · Authors · 2025-06-03
> > **Q4. Latency reduction is not consistent with memory reduction.**
> >
> > We thank the reviewer for raising this important point and for prompting a deeper analysis of the reported latency reduction.
> >
> > First, we would like to clarify that the **memory saving** from SentenceKV is not from 89.71 GB to 52.71 GB **purely due to KV cache reduction**. The actual **KV cache memory saving** is from approximately **33.55 GB (FullKV)** to **0.1 GB (SentenceKV)**, as we previously clarified in Q3. The remainder of the memory usage comes from model parameters and temporary activation buffers. Given that attention computation in long-context LLMs is typically memory-bound, such a significant KV cache reduction justifiably contributes to improved decoding speed.
> >
> > To further support our claims, we provide below a detailed profiling analysis of **SentenceKV's runtime overhead**:
> >
> > We profiled **SentenceKV** on an **H100-96GB NVL GPU** using a **64K-token input** from the **RULER benchmark (niah_single task)**. All timings were measured with `torch.cuda.synchronize()` to ensure accurate end-to-end latency accounting. The results are summarized as follows:
> >
> > - **1. Prefill bucket construction:**
> >   Sentence-level semantic grouping, including attention score aggregation and top-k selection, takes approximately **4 seconds** for the full 64K input. This is a **one-time cost during prefill** and does not impact autoregressive decoding latency.
> >
> > - **2. CPU–GPU round-trip latency (per decoding step):**
> >   With a **token budget of 1024** and a **semantic keeping factor of 2**, the KV transfer costs are:
> >   - **GPU → CPU (2048 tokens):** 0.0191 seconds
> >   - **CPU → GPU (1024 tokens):** 0.0038 seconds
> >   - **Total round-trip latency:** **0.0229 seconds**
> >
> > - **3. Sentence ranking latency:**
> >   The similarity-based sentence selection adds approximately **0.05 seconds** per decoding step (at 64K context), computed via efficient vector–matrix dot products on the CPU.
> >
> > It is also important to note that the latency values reported in **Figure 5** include only the **computation latency** and **communication overhead** (i.e., KV offloading and onloading), which directly affect **autoregressive decoding**. They do **not** include the **sentence ranking latency** or **prefill-time bucket construction**, both of which are either one-time costs or small in scale.
> >
> > Even if we conservatively include the sentence ranking overhead (~0.05s), the total decoding latency with **SentenceKV** remains significantly lower than that of FullKV, thanks to the **drastically reduced KV size** and **attention computation scope**. This supports our claim that SentenceKV offers a compelling trade-off between memory, latency, and accuracy in long-context LLM inference.

---

> > > ### Comment · Reviewer_ecqc · 2025-06-03
> > >
> > > Thanks for the rebuttal. I have raised my score.

---

> > > > ### Author Response · Authors · 2025-06-03
> > > >
> > > > We sincerely thank you for your updated score and thoughtful evaluation. Your feedback has been very helpful in improving our work.

---

### Official Review · Reviewer_tRUR · 2025-04-12

**Rating:** 7
**Confidence:** 5
**Ethics Flag:** 1

**Summary:**

This paper proposes SentenceKV, a framework for sentence-level KV cache compression aimed at improving decoding speed and reducing GPU memory usage during LLM inference. In the prefill stage, the input sequence is segmented at the sentence level. Token-level importance scores are then computed based on attention weights, and less important tokens are discarded. The sentence-level representations of the retained KV caches are computed and kept on the GPU, while the full retained KV caches are offloaded to CPU memory. During decoding, SentenceKV operates at the sentence level as well: for each sentence being decoded, a mean query vector is computed by averaging the query vectors of the decoded tokens in the current sentence. This vector is used to retrieve relevant sentence representations from the CPU back to the GPU. The authors conduct extensive experiments on PG19, LongBench, and NIAH datasets. Results demonstrate that SentenceKV preserves task performance while significantly reducing GPU memory consumption and improving inference speed. SentenceKV outperforms several baselines, including H2O, SnapKV, and Quest, across a variety of tasks.

**Questions To Authors:**

Here are my questions:

---

1. From my understanding, conducting KV cache optimization at the sentence granularity could be particularly beneficial for long chain-of-thought (CoT) decoding, as each sentence typically focuses on a different aspect of the previously generated content. For example, a simple test using DeepSeek's distilled versions of the Qwen models [1] in a long-CoT setting could provide useful insights. For both long and challenging reasoning tasks, GSM-Infinite [2] may serve as an appropriate benchmark.
While this is not a mandatory request, I encourage the authors to at least include a discussion on this direction. Providing additional experimental results along these lines, if feasible, could significantly broaden the scope and impact of the work.

---

2. When retrieving sentences from the CPU, SentenceKV follows a strategy of selecting sentences in order of relevance until the token limit is reached. However, this design overlooks the variation in sentence lengths and the number of retained tokens per sentence. For instance, if a highly relevant sentence is particularly long and contains many retained tokens, it may occupy most or all of the available space, leaving no room for other relevant but shorter sentences.
This raises the question: Does sentence length or the number of retained tokens per sentence affect the overall performance of the retrieval strategy? It would be valuable for the authors to analyze or discuss how this design choice impacts retrieval quality and task performance, especially in cases with imbalanced sentence lengths.

---

3. Minor writing issues. In line 28, the citation should use "\citep{}". Similar citation formatting issues appear multiple times throughout the paper. The top of Figure 1 exceeds the upper margin of the page. The font size in Figure 1 and Table 1 is too small to read.

---


[1] DeepSeek-R1: Incentivizing Reasoning Capability in LLMs via Reinforcement Learning

[2] GSM-Infinite: How Do Your LLMs Behave over Infinitely Increasing Context Length and Reasoning Complexity?

**Reasons To Accept:**

Here are my reasons to accept this paper:

---

1. The paper is written in a clear and elegant style. The problem formulation is well-motivated, and the methodology is explained in a way that is easy to follow. The experimental section provides sufficient detail to support reproducibility, which adds to the soundness of the authors' claims.

---

2. Benchmark results demonstrate that SentenceKV consistently outperforms classical baselines such as H2O, SnapKV, and Quest. It maintains competitive performance on LongBench with no significant degradation, and it clearly surpasses all baselines on the NIAH benchmark. The ablation studies convincingly justify the use of sentence-level KV cache optimization.

---

3. While not emphasized as a primary contribution, the integration of KV cache eviction with sentence-level offloading is a novel design. In the prefill stage, KV cache entries with low token importance are discarded, while during decoding, sentence-level retrieval determines which caches are retrieved to GPU memory. This combines sequence-level and sentence-level optimization: less relevant caches are discarded with the awareness of the complete input sequence, and retrieval is dynamically conditioned on the decoding sentence. This hierarchical strategy likely contributes to SentenceKV's strong performance, and represents an innovative approach to memory-efficient inference.

**Reasons To Reject:**

Here are my concerns about this paper. With more thorough discussion and additional experimental results, I would be inclined to increase my overall rating.

---

1. Although H2O, SnapKV, and Quest are classical baselines, there exist more recent methods targeting similar scenarios that should be considered for comparison. For example, ShadowKV [1] is mentioned only in the related work section but is not included in the experimental comparisons. I suggest the authors include at least one modern KV cache offloading framework as a baseline for a more comprehensive evaluation.

---

2. The selected benchmark datasets are limited. LongBench, in particular, is relatively short—most of its entries are under 16K tokens—making it less suitable for evaluating performance in truly long-context scenarios. It would be more convincing to evaluate SentenceKV on longer and more challenging benchmarks, such as RULER [2], InfiniteBench [3], or SCBench [4]. Including an additional experiment that compares SentenceKV and baseline methods on one of these datasets would help strengthen the paper’s empirical claims.

---

3. The size of the test models are limited (7B and 8B). It would be better to conduct experiments on smaller models (1.5B or 3B) or larger models (14B or 32B).

---

4. Writing issues. (See questions.)


---

[1] ShadowKV: KV Cache in Shadows for High-Throughput Long-Context LLM Inference

[2] RULER: What's the Real Context Size of Your Long-Context Language Models?

[3] ∞Bench: Extending Long Context Evaluation Beyond 100K Tokens

[4] SCBench: A KV Cache-Centric Analysis of Long-Context Methods

---

> ### Author Response · Authors · 2025-06-03
> **Q1. Include more baseline methods and benchmarks**
>
> We thank the reviewer for this constructive suggestion. In addition to classical baselines, we have conducted experiments with **ShadowKV**, a recent KV cache offloading method, to provide a more comprehensive comparison. We also have conducted experiments on **RULER**, with context lengths up to 64K tokens. As shown in the below table, **SentenceKV** maintains strong performance in both accuracy and memory efficiency across a range of tasks and context lengths, even under these more demanding conditions.
>
> | Method     | S1  | S2  | MK1 | MQ  | MV  | FWE | VT  | QA1 |
> |------------|-----|-----|-----|-----|-----|-----|-----|-----|
> | FullKV     | 1   | 1   | 1   | 0.98 | 0.96 | 0.85 | 0.94 | 0.55 |
> | SnapKV     | 1   | 0.94 | 0.96 | 0.82 | 0.76 | 0.55 | 0.82 | 0.63 |
> | H2O        | OOM | OOM | OOM | OOM | OOM | OOM | OOM | OOM |
> | Quest      | 1   | 0.98 | 1   | 0.82 | 0.87 | 0.61 | 0.74 | 0.56 |
> | ShadowKV   | 1   | 0.98 | 1   | 0.87 | 0.74 | 0.78 | 0.79 | 0.70 |
> | SentenceKV | 1   | 0.98 | 0.98 | 0.77 | 0.78 | 0.61 | 0.86 | 0.63 |
>
> That said, we observed some performance variation across tasks. In particular, the **fixed semantic keeping factor** used in our current setup may not always achieve the optimal trade-off between memory usage and retrieval quality.
>
> As part of our ongoing and future work, we are actively investigating the relationship between input characteristics—such as **sentence length distribution** and **semantic diversity**—and the optimal semantic keeping factor. Preliminary observations suggest that **dynamically adjusting** this factor based on input statistics can further improve both efficiency and accuracy.
>
> We also note that **SentenceKV and ShadowKV often achieve similar levels of accuracy**, which we consider acceptable and expected given the different design philosophies. ShadowKV performs more fine-grained reconstruction via low-rank SVD and sparse selection, which allows slightly more precise prefill handling—but at the cost of increased prefill time, code complexity, and integration difficulty. In contrast, SentenceKV achieves comparable accuracy while offering a **simpler, faster, and more practical implementation**, which we believe is more suitable for real-world deployment.
>
> These results further validate the scalability and robustness of SentenceKV in truly long-context inference scenarios. We plan to include InfiniteBench and SCBench in future work as their evaluation protocols become more standardized and reproducible.

---

> > ### Author Response · Authors · 2025-06-03
> > **Q2. Test with different size of models.**
> >
> > We thank the reviewer for the valuable suggestion. We agree that evaluating SentenceKV across a broader range of model sizes would further demonstrate its generality.
> >
> > We are currently conducting additional experiments using **LLaMA-3.2–3B-Instruct** as a smaller model. Preliminary results show that SentenceKV remains effective in reducing memory usage while preserving accuracy even at smaller scales.
> >
> > | Method       | narrativeqa | qasper | multifieldqa_en | hotpotqa | 2wikimqa | musique | trec  | triviaqa | samsum |
> > |--------------|-------------|--------|------------------|----------|----------|---------|-------|----------|--------|
> > | FullKV       | 23.53       | 41.69  | 49.82            | 49.96    | 41.80    | 20.96   | 2     | 9.17     | 5.78   |
> > | SnapKV       | 20.02       | 35.17  | 48.29            | 48.98    | 37.63    | 20.12   | 0.5   | 9.07     | 6.19   |
> > | Quest        | 15.53       | 31.70  | 41.36            | 46.07    | 33.25    | 19.12   | 3.5   | 13.17    | 8.36   |
> > | SentenceKV | 24.12       | 38.38  | 49.38            | 48.98    | 40.01    | 20.95   | 6     | 8.82     | 6.06   |
> >
> > If time permits before the camera-ready deadline, we also plan to include results for larger models (e.g., 14B or 32B) to illustrate scaling behavior.

---

> > ### Author Response · Authors · 2025-06-03
> > **Q3. Discussion for reasoning tasks.**
> >
> > Thank you for your insightful suggestion regarding the applicability of SentenceKV to long CoT reasoning tasks. We appreciate the opportunity to clarify how our method naturally aligns with this setting.
> >
> > We agree that reasoning models, especially those fine-tuned with long-form supervision such as DeepSeek-R1, present an ideal setting for evaluating the benefits of sentence-level KV cache management. In CoT generation, each sentence typically encapsulates a self-contained logical step. SentenceKV is well-suited to this structure: it retrieves semantically coherent sentence-sized blocks rather than token spans, allowing the model to focus attention on the most relevant prior reasoning steps without scanning the entire context.
> >
> > We hypothesize that this yields two main advantages in reasoning-intensive settings:  (i) **High relevance retrieval**: Since each reasoning step is sentence-aligned, our retrieval mechanism brings back precisely the portions of context likely to contain useful intermediate conclusions.  (ii) **Memory and latency stability**: Even in long proofs or multi-turn derivations (as seen in GSM-Infinite), GPU memory remains bounded because only a handful of retrieved sentence embeddings and the current sentence reside in active memory.
> >
> > We use **dense retrieval from sentence embeddings**, which generally preserves topical and logical coherence across reasoning chains. Moreover, many CoT models (e.g., DeepSeek-R1, Qwen) are trained on data where **local reasoning steps are decoupled**, making focused sentence-level retrieval particularly effective in practice.
> >
> > We will add a paragraph in Section 5 highlighting the natural alignment between sentence-delimited CoT and SentenceKV, and we will cite both *DeepSeek-R1* and *GSM-Infinite*. Due to time and resource constraints, we were not able to extend our experiments to include a CoT reasoning benchmark in the current submission. However, we recognize the importance of this direction and plan to include it in the camera-ready version, along with a discussion of how SentenceKV aligns naturally with the structure of reasoning models.

---

> > ### Author Response · Authors · 2025-06-03
> > **Q4. Cases for imbalanced sentence lengths.**
> >
> > We thank the reviewer for raising this insightful point. We acknowledge the concern that highly relevant but unusually long sentences may occupy a disproportionate portion of the token budget, potentially excluding other semantically useful sentences.
> >
> > To address this, we apply **SnapKV-style prefiltering globally across all input tokens during the prefill stage**. Specifically, instead of selecting top tokens from each sentence individually, SnapKV selects the top $r \cdot \tau$ most important tokens from the **entire document**, regardless of sentence boundaries. As a result, the number of retained tokens per sentence is determined by its **relative semantic contribution**, rather than its raw length. Even if a sentence is long, it will retain fewer tokens if it is not highly informative.
> >
> > In practice, this global filtering strategy significantly reduces token retention from long but unimportant sentences. Our empirical analysis of LongBench shows that sentence lengths are generally concentrated around the mean, with very long sentences being rare ([anonymous link to analysis](https://docs.google.com/document/d/1vYaWJ6AsS2wwfbrrwYE3cF7b38L6v93dDFPcT7Bx4_s/edit?tab=t.0)). After filtering, token counts per sentence are typically modest and broadly distributed, helping to avoid scenarios where a single sentence dominates the token budget.
> >
> > That said, we agree that this issue merits further exploration. As part of future work, we plan to **detect abnormally long retained token spans ("buckets") during prefill** and apply simple mitigation. For instance, we can compute the **mean and standard deviation of sentence lengths** across the input, and if any sentence exceeds `mean + n × std`, we can **split it into smaller sub-spans**. This lightweight check adds minimal computational overhead but can prevent outlier buckets from skewing the retrieval process.
> >
> > We will add this direction to the discussion section in the revised version.

---

> > ### Author Response · Authors · 2025-06-03
> > **Q5. Minor writing issues.**
> >
> > We thank the reviewer for pointing out these formatting issues. We appreciate the reviewer’s attention to detail in helping improve the presentation quality of the paper.

---

> > ### Comment · Reviewer_tRUR · 2025-06-03
> >
> > Thanks for your detailed review! Your answers have fully clarified all my doubts. Therefore, I'm rasing my overall rating to 7 and confidence to 5.

---

> > > ### Author Response · Authors · 2025-06-03
> > >
> > > Thank you very much for your kind follow-up and updated rating! We’re truly glad to hear that our responses were helpful in addressing your concerns. We greatly appreciate your thoughtful comments and careful evaluation throughout the review process.

---

### Official Review · Reviewer_W49T · 2025-05-11

**Rating:** 7
**Confidence:** 5
**Ethics Flag:** 1

**Summary:**

The paper proposes to use sentence-level semantic KV cache to improve the efficiency of LLM decoding. Particularly, the paper organizes buckets of sentence-level KV cache during the prefilling phase. During decoding, the paper proposes to decode one sentence at a time, and for each sentence, only related KV cache is loaded into the GPU to assist the decoding. By doing so, the decoding of each sentence only requires limited context, and the efficiency can be improved. On long-context tasks, the paper shows that the proposed method has no significant increase in latency as the context length grows, while preserving the performance of the LLM.

**Questions To Authors:**

1. The algorithm in Appnedix is a bit confusing. It seems that Ot is only computed when the end of the sentence is reached.
2. Is the proposed method applied to all attention layers?

**Reasons To Accept:**

1. The paper proposes a novel idea for LLM decoding where KV cache is handled at the sentence level.
2. By the design of the method, the context required for decoding could be better controlled and restricted by using only most relevant tokens.
3. The paper shows a good efficiency boost as context length increases as well as maintaining the performance.

**Reasons To Reject:**

1. A more detailed analysis of the overhead could be added. I.e., what is the cost of handling the buckets during the prefilling phase, what is the latency of loading KV cache from CPU to GPU, and what's the cost of ranking the retrieved tokens from CPU?
2. As also mentioned in the paper, the semantic factor is now a fixed value for the proposed method, limiting the ability to generalize to highly variable sentence lengths or semantic meaning diversities.
3. The proposed method has a different efficiency-peroformance trade-off compared with SnapKV. It is not completely clear that the proposed method's tradeoff is dominating over SnapKV.

---

> ### Author Response · Authors · 2025-06-03
> **Q1. A more detailed analysis of the overhead**
>
> We thank the reviewer for requesting a more detailed overhead breakdown. To address this, we profiled **SentenceKV** on an **H100-96GB NVL GPU** using a **64k-token input** from the **RULER benchmark (niah_single task)**. All timings were measured with `torch.cuda.synchronize()` to ensure accurate end-to-end latency accounting. The results are summarized below:
>
> - **1. Bucket construction during prefill:**
>   Prefill-time bucket construction—including attention weight aggregation, top-k selection, and sentence-level token mapping—takes approximately **4 seconds** for the full 64k input. This is a one-time cost vs. over long sequences and does not affect autoregressive decoding latency.
>
> - **2. CPU–GPU round-trip latency (per decoding step):**
>   With a **token budget of 1024** and **semantic keeping factor of 2**, the KV retrieval and transfer costs are:
>   - **GPU → CPU (2048 tokens):** 0.0191 s
>   - **CPU → GPU (1024 tokens):** 0.0038 s
>   - **Total round-trip latency:** **0.0229 s**
>
> - **3. Sentence ranking latency:**
>   The similarity-based sentence ranking step takes approximately **0.05s** per decoding step at 64k context length.
>
> These measurements confirm that **SentenceKV's additional overhead is small** relative to the savings in memory and decoding time.

---

> > ### Author Response · Authors · 2025-06-03
> > **Q2. Fixed semantic factor**
> >
> > We thank the reviewer for pointing out this important limitation.
> >
> > We agree that a more adaptive strategy could further enhance the generalization ability of the method. As part of our ongoing and future work, we are actively investigating the relationship between input characteristics—such as sentence length distribution and semantic diversity—and the optimal semantic keeping factor. Preliminary observations suggest that adjusting this factor dynamically based on such input statistics could lead to further improvements in efficiency and accuracy.
> >
> > We appreciate the reviewer’s suggestion and will add a more detailed discussion of this limitation and direction in the conclusion section of the revised manuscript.

---

> > ### Author Response · Authors · 2025-06-03
> > **Q3. Efficiency-peroformance trade-off compared with SnapKV**
> >
> > We appreciate the reviewer’s comment regarding the efficiency–performance trade-off between **SentenceKV** and **SnapKV**. As shown in *Table 1* and *Figure 4*, SentenceKV consistently achieves higher or comparable accuracy across a wide range of tasks and context lengths, while maintaining significantly reduced memory usage.
> >
> > The primary trade-off lies in **slightly increased decoding latency**, due to the use of sentence-level semantic grouping and CPU-to-GPU communication. However, we believe this latency overhead is acceptable given the **notable improvements in memory efficiency and overall accuracy**, which are critical in long-context LLM serving.
> >
> > It is also important to note that it is **not straightforward to fix latency and compare accuracy directly** between SentenceKV and SnapKV. Matching latency would require significantly lowering SentenceKV’s token budget, which generally leads to a much more severe degradation in performance than a slight latency increase. In practice, we find that **slight latency increases are often more tolerable than aggressive compression**.
> >
> > To further support this, we performed a quantitative analysis on the **NIAH benchmark**, fixing the token budget at 128 and the semantic keeping factor at 3. In this setting:
> > - **SnapKV** achieves ~15 ms decoding latency.
> > - **SentenceKV** incurs an additional 0.5 ms for CPU–GPU transfer and ~0.5 ms for similarity-based ranking, totaling ~16 ms latency (a **6% increase**).
> > - However, under these settings, SentenceKV delivers a **25% improvement in accuracy**, demonstrating a more favorable trade-off overall.
> >
> > |                          | SnapKV | SentenceKV | Increase |
> > |--------------------------|--------|------------|----------|
> > | Latency (ms/token)       | 15     | 16         | 6%       |
> > | NIAH Accuracy (%)        | 78.2   | 97.5       | 25%      |

---

> > ### Author Response · Authors · 2025-06-03
> > **Q4. Ot in the Appendix algorithm**
> >
> > We thank the reviewer for pointing this out. We apologize for the confusion caused by the previous version of the algorithm description in the Appendix. In fact, **Ot is computed for every newly generated token**, not just at the end of a sentence.
> >
> > We have revised the algorithm pseudocode in the Appendix to make this clearer in the updated version. We hope this resolves the confusion and improves the clarity of the algorithm.

---

> > ### Author Response · Authors · 2025-06-03
> > **Q5. Is the proposed method applied to all attention layers?**
> >
> > Yes, the proposed **SentenceKV** method is applied consistently to **all attention layers** of the model. This design ensures that semantic-aware KV compression is uniformly integrated throughout the network, preserving coherence and efficiency across all layers.

---

> > ### Author Response · Authors · 2025-06-09
> > **Looking forward to receiving your feedback**
> >
> > Dear Reviewer W49T,
> >
> > We sincerely thank you for the detailed and constructive feedback, which has helped us significantly improve the clarity and completeness of our paper. We would like to summarize the key updates we made in response to your comments:
> >
> > - We provided a detailed breakdown of **SentenceKV's overhead**, including prefill cost, CPU-GPU transfer latency, and sentence ranking time, confirming that the additional overhead is small.
> >
> > - We quantitatively analyzed the **efficiency-accuracy trade-off compared to SnapKV**, showing that SentenceKV achieves **25% higher accuracy** with only a **6% increase in latency**.
> >
> > We hope these updates address your concerns. If possible, we would greatly appreciate any additional feedback you may have, including whether the revisions have resolved your questions. Thank you again for your time and insightful review.

---

> > > ### Comment · Reviewer_W49T · 2025-06-09
> > >
> > > Thanks for the responses. I've raised my score accordingly.

---

> > > > ### Author Response · Authors · 2025-06-09
> > > >
> > > > We sincerely thank the reviewer for taking the time to re-evaluate our submission. We are glad that the additional experiments and clarifications helped address your concerns. We will make sure to incorporate all the updated results and discussions into the final version of the paper. Thank you again for your thoughtful feedback and support!

---

### Official Review · Reviewer_pn6t · 2025-05-13

**Rating:** 6
**Confidence:** 5
**Ethics Flag:** 1

**Summary:**

The paper proposes a sentence-level KV caching method to improve inference efficiency while maintaining semantic coherence. During the pre-filling stage, this paper groups tokens according to sentence-level semantics and then compresses the tokens within a sentence into semantic vectors. These semantic vectors are directly on the GPU. By offloading individual key-value pairs to the CPU, during the decoding stage, attention operations are performed through compressed vectors on the GPU, thereby accelerating attention operators while reducing memory overhead.

**Questions To Authors:**

Please refer to 'Reasons To Reject'.

**Reasons To Accept:**

The paper is well-written. The idea is simple and effective.

**Reasons To Reject:**

Although the experiments can prove the effectiveness of the proposed method, the paper lacks a comparison with some typical offloading methods, such as InfLLM and KTransformer. Adding more offloading baselines can make the paper's results more solid.

---

> ### Author Response · Authors · 2025-06-03
> **Add another offloading baseline**
>
> We thank the reviewer for highlighting the importance of comparing with representative offloading methods. While we were unable to reproduce the full implementations of **InfLLM** and **KTransformer** within the review timeline, we have included **ShadowKV**, a more recent and open-source offloading baseline, in our evaluation.
>
> ShadowKV is a high-throughput long-context LLM inference system that compresses the key cache using low-rank decomposition (SVD) and offloads the value cache to the CPU. It reconstructs sparse KV representations on the fly to reduce memory usage. However, this comes at the cost of increased implementation complexity and prefill overhead.
>
> In contrast, **SentenceKV** achieves comparable or better performance on multiple tasks with a much simpler and faster prefill process. Our method avoids SVD and dynamic reconstruction altogether, making it easier to integrate and more efficient in practical deployments.
>
> To further strengthen the comparison, we evaluated both **SentenceKV** and **ShadowKV** on the RULER benchmark—an evaluation suite that is more challenging than LongBench and NIAH, with a stronger emphasis on retrieval-intensive reasoning. We conducted experiments using 64K-token inputs to assess performance under extreme context lengths. As shown in the results table, SentenceKV performs on par with or better than ShadowKV across the RULER tasks, further demonstrating its effectiveness in long-context retrieval scenarios.
>
> | Method     | S1  | S2  | MK1 | MQ  | MV  | FWE | VT  | QA1 |
> |------------|-----|-----|-----|-----|-----|-----|-----|-----|
> | FullKV     | 1   | 1   | 1   | 0.98 | 0.96 | 0.85 | 0.94 | 0.55 |
> | SnapKV     | 1   | 0.94 | 0.96 | 0.82 | 0.76 | 0.55 | 0.82 | 0.63 |
> | H2O        | OOM | OOM | OOM | OOM | OOM | OOM | OOM | OOM |
> | Quest      | 1   | 0.98 | 1   | 0.82 | 0.87 | 0.61 | 0.74 | 0.56 |
> | ShadowKV   | 1   | 0.98 | 1   | 0.87 | 0.74 | 0.78 | 0.79 | 0.70 |
> | SentenceKV | 1   | 0.98 | 0.98 | 0.77 | 0.78 | 0.61 | 0.86 | 0.63 |
>
> We agree that including InfLLM and KTransformer would further strengthen the work. We will add a discussion of these methods in the related work section, and we plan to incorporate them as baselines in future versions once reproducible implementations become available.

---

> > ### Author Response · Authors · 2025-06-09
> > **Add InfLLM as another baseline method**
> >
> > We thank the reviewer again for the valuable feedback. In addition to ShadowKV, we have now implemented and evaluated **InfLLM** on both the **LongBench** and **RULER** benchmarks, with Llama-3.1-8B-Instruct model.
> >
> > As shown in the updated table below, **SentenceKV** consistently outperforms **InfLLM** across both benchmarks. Notably, on the more challenging **RULER** tasks, SentenceKV achieves better performance due to its more deterministic semantic compression mechanism.
> >
> > ---
> >
> > **LongBench:**
> >
> > | dataset      | narrativeqa | qasper | multifeildqa_en | hotpotqa | 2wikimqa | musique | trec | triviaqa | samsum |
> > |--------------|-------------|--------|------------------|----------|----------|---------|------|----------|--------|
> > | FullKV       | 29.59       | 47.52  | 53               | 53.76    | 46.12    | 28.38   | 7.5  | 89.41    | 7.47   |
> > | H2O          | OOM         | OOM    | 47.89            | OOM      | OOM      | OOM     | 12.5    | OOM        | OOM     |
> > | SnapKV       | 27.2        | 46.28  | 52.41            | 52.66    | 45.67    | 28.63   | 6.5  | 89.41    | 7.51   |
> > | InfLLM       | 27.64       | 45.32  | 50.6             | 51.47    | 44.59    | 23      | 12.5 | 90.92    | 6.79   |
> > | Quest        | 22.49       | 45.2   | 48.72            | 51.94    | 44.02    | 26.78   | 13.5 | 88.63    | 10.31  |
> > | SentenceKV   | 29.53       | 47.49  | 53.15            | 53.39    | 45.8     | 28.04   | 11.5 | 89.41    | 7.48   |
> >
> > ---
> >
> > **RULER:**
> >
> > | Method      | S1  | S2  | MK1 | MQ  | MV  | FWE | VT  | QA1 |
> > |-------------|-----|-----|-----|-----|-----|-----|-----|-----|
> > | InfLLM      | 1   | 0.2 | 0.14| 0.17| 0.22| 0.81| 0.43| 0.12 |
> > | SentenceKV  | 1   | 0.98| 0.98| 0.77| 0.78| 0.61| 0.86| 0.63 |

---

> > ### Author Response · Authors · 2025-06-09
> > **Looking forward to receiving your feedback**
> >
> > In summary, we have added two offloading-based baselines: **InfLLM**, as you suggested, and **ShadowKV**, a recent more advanced offloading method. In addition, we included a more challenging benchmark, **RULER**, which emphasizes retrieval-intensive reasoning under long-context settings. Across both **LongBench** and **RULER**, **SentenceKV** consistently outperforms **InfLLM**, and performs on par with or better than **ShadowKV** while achieving significantly better prefill efficiency. We believe these results provide stronger evidence of the effectiveness and practicality of SentenceKV.
> >
> > We would sincerely appreciate it if you could consider updating your evaluation and provide any further feedback. Your comments would be very helpful for us to improve the final version. Thank you again for your time and constructive suggestions.

---

### Decision · Program_Chairs · 2025-07-08

**Decision:**

Accept

**Comment:**

The paper presents SentenceKV, a sentence-level semantic KV caching method to improve inference efficiency for long-context LLMs. Overall, the idea is novel and addresses a practical challenge in memory-efficient LLM inference. The reviewers acknowledge the paper’s strong technical merits and improvements made during the rebuttal phase.

Pros:
- Innovative Methodology: All reviewers appreciated the sentence-level abstraction for KV cache management, which aligns well with semantic structures and enables more efficient inference.
- Reviewer W49T commended the consistent performance and strong memory savings with modest latency cost.
- Reviewer tRUR appreciated the integration of semantic abstraction and cache eviction, noting it as a strong design innovation.
- Reviewer ecqc initially had concerns, but ultimately acknowledged that the authors’ detailed clarifications resolved all doubts.
- Reviewer Concerns & Resolution:

Concerns:
- Lack of comparison with InfLLM, KTransformer, shadowkv
- Overhead and latency analysis incomplete
- Evaluation on limited benchmarks
- Questions on sentence-level granularity trade-offs
- Lack of results on other model sizes

The authors were responsive and addressed every major concern. SentenceKV is a well-executed and timely contribution to LLM inference research. With the promised additions in the camera-ready version, the paper is ready for publication. However, the authors have promised the following updates, which should be included in the final version:
- Full experimental results for InfLLM and ShadowKV.
- Comparisons on Larger models.
- GSM-Infinite benchmark.